# A Mini Review on Sewage Sludge and Red Mud Recycling for Thermal Energy Storage

Yaxuan Xiong [1,*], Aitonglu Zhang [1], Yanqi Zhao [2,*], Qian Xu [3] and Yulong Ding [4]

[1] Beijing Key Lab of Heating, Gas Supply, Ventilating and Air Conditioning Engineering, Beijing University of Civil Engineering and Architecture, Beijing 100044, China; 2108140422022@stu.bucea.edu.cn

[2] Jiangsu Key Laboratory of Process Enhancement and New Energy Equipment Technology, School of Energy Science and Engineering, Nanjing Tech University, Nanjing 211816, China

[3] School of Energy and Environmental Engineering, University of Science and Technology Beijing, Beijing 100083, China; qianxu@ustb.edu.cn

[4] Birmingham Center for Energy Storage, University of Birmingham, Birmingham B15 2TT, UK; y.ding@bham.ac.uk

* Correspondence: xiongyaxuan@bucea.edu.cn (Y.X.); hazhaoyq@126.com (Y.Z.)

**Abstract:** Sewage sludge and red mud, as common industrial waste, have become a research hotspot in the field of achieving carbon peaking and carbon neutrality, reducing carbon emissions, and solving environmental problems. However, their treatment and disposal have always been a difficult problem in the environmental field. Utilizing these two materials for thermal energy storage can not only improve energy utilization efficiency but also further reduce carbon emissions during their treatment process, providing a new approach for sustainable development in the industrial sector. This article summarizes the research progress for the resource recovery of sewage sludge and red mud for direct thermal energy recovery and composite phase change energy storage. After proper treatment, sludge and red mud can be directly used as energy storage materials. In addition, sludge and red mud can be combined with phase change materials to prepare composite materials with an excellent energy storage performance. This composite has broad application prospects in fields such as solar energy utilization and building energy efficiency. However, there are still some challenges and issues in this resource recovery and utilization, such as potential environmental pollution during the treatment process, the long-term stability of energy storage materials, and cost-effectiveness, which require further research and resolution. The purpose of this paper is to evaluate the potential of sewage sludge and red mud as energy storage materials, to explore their feasibility and advantages in practical applications, and to reveal the research progress, technical challenges, and future development directions of these two materials in the field of thermal energy storage.

**Keywords:** sewage sludge; red mud; thermal energy storage; solid waste resource utilization; energy recovery; carbon reduction

## 1. Introduction

With the rapid development of industrialization, the production of industrial sludge, as a major solid waste, has shown a rapid growth trend, and large quantities of industrial sludge need to be properly disposed of [1]. Sewage discharge has increased dramatically, with sewage sludge (SS) inevitably generated [2]. SS refers to the solid or semi-solid residues generated during the wastewater treatment process, and these residues are primarily composed of microbial cells, incompletely decomposed organic matter, inorganic particles, and colloids, among other substances, resulting in a highly complex composition, which not only contains nutrients such as organic matter, nitrogen, phosphorus, and potassium, but also contains harmful substances like heavy metals, pathogens, and parasite eggs [3]. Therefore, the treatment and disposal of SS is not only crucial to the operational efficiency of wastewater treatment plants but also directly impacts environmental safety and human

health [4,5]. In addition, the aluminum industry produces a kind of industrial solid waste red mud (RM) when extracting alumina [6]. RM is a harmful alkaline waste residue discharged during the refining of bauxite into alumina, and 1 to 1.5 tons of RM is discharged with every ton of alumina manufactured [7]. RM contains a large amount of ferric oxide, which gives it a unique red color, and the accumulation of RM is increasingly causing environmental pollution [8]. Moreover, regular maintenance is required for sedimentary RM in dams or embankments, resulting in a cost increase of 7 to 15 USD/ton, which indirectly raises the production cost of aluminum by up to 5% [9]. In the EU, the annual sludge solids production is about 7.2 million tons, and in China, the sludge production reaches 39 million tons/year [10]. It is estimated that, globally, each person generates an average of 35 to 85 g of dry solid sludge per day [11]. The global annual production of RM exceeds 150 million tons [12], and the global production of activated sludge (with a water content of 80%) is estimated to reach 103 million tons by 2025 [13]. As shown in Figure 1, SS and RM that are not properly treated can cause serious pollution and harm to the environment, water resources, soil, atmosphere and human health [14]. To eliminate these hazards, a range of treatment and disposal measures are required. As shown in Figure 2, eliminating the hazards of SS and RM requires comprehensive measures, including direct waste treatment, developing new utilization methods, and reducing the generation of waste from the source. SS is treated in a variety of ways, including compression and dewatering, incineration, composting, biological treatment, and land use [15]. Each of these methods has its own characteristics and aims to achieve sludge reduction, stabilization, harmlessness, carbon reduction, and resource utilization [16]. Particularly, biological treatment is a natural method of degrading organic substances through the metabolic reaction of microorganisms, which has the advantages of reuse and long-term stability [17,18]. In the context of a circular economy, the proper thermochemical treatment of sludge, combined with energy generation and resource recovery processes, or the use of thermochemical residues as fertilizer, seems to be a prudent and practical strategy [19]. To fit the concept of the circular economy and provide a long-term viable solution for the application of sewage sludge, heat treatment methods such as combustion and gasification have been considered as an alternative [20]. In addition, the reuse of SS can reduce the emission of greenhouse gases such as carbon dioxide and nitrous oxide [21]. The treatment of RM is primarily focused on its resource utilization. However, the high content of alkaline substances in RM is the main obstacle to its large-scale resource utilization [22]. In recent years, studies have shown that the utilization rate of RM is gradually increasing through de-alkalinization and engineering applications [7,23,24]. RM has an application potential in the field of engineering, environmental protection, and the extraction of metal elements, such as roadbed engineering and filling engineering, and other large-scale engineering projects may become an effective means of RM resource utilization, which could realize the cost-effective utilization of RM [25]. However, the resource utilization ratio of industrial sludge is very low, while the resource utilization ratio of RM is less than 10% [26]. Carbon emission reduction, as an important goal of global environmental protection, requires the resource utilization of various industrial sludges [27]. Converting the organic matter in the sludge into energy or resource products through technical means can not only reduce greenhouse gas emissions, but also realize the resource utilization of waste, which has significant environmental protection and economic benefits [28,29]. Wang et al. [27] recovered large amounts of carbon from sludge through acidogenic fermentation and performed denitrification, which helps to reduce greenhouse gas emissions. Huang et al. [30] studied sludge-derived concrete and found that the sludge treatment method had the potential to achieve an 83.48% reduction in carbon emissions compared to conventional cement. Meanwhile, energy demands require us to increase the exploration of energy storage technology, and thermal energy storage (TES) technology has attracted extensive attention as an efficient and environmentally friendly means of energy utilization, which can further reduce the mismatch between the energy supply and human energy demand [31]. TES mainly includes sensible heat storage, latent heat storage, and thermochemical heat storage [32]. Figure 3 illustrates the applications of the three TES

technologies and different types of TES that SS- and RM-processing technologies can be attributed to.

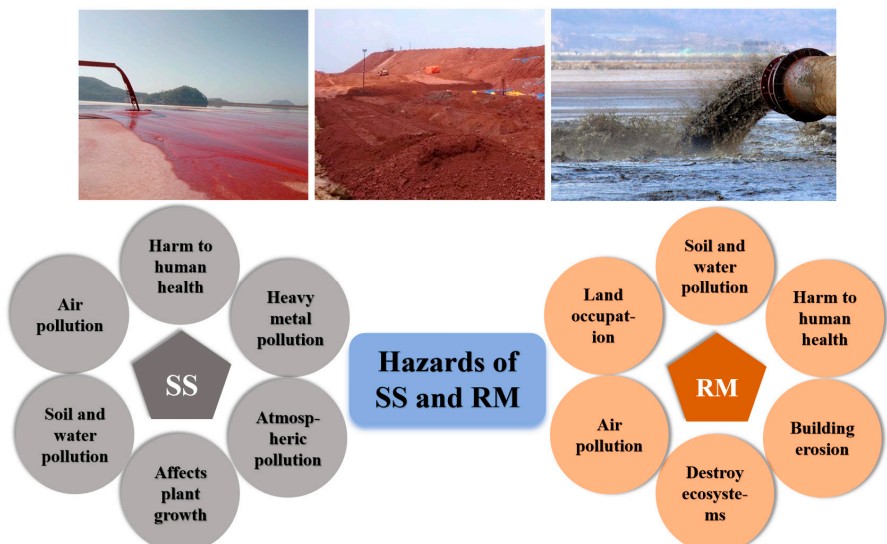

**Figure 1.** Hazards of SS and RM [8,33,34].

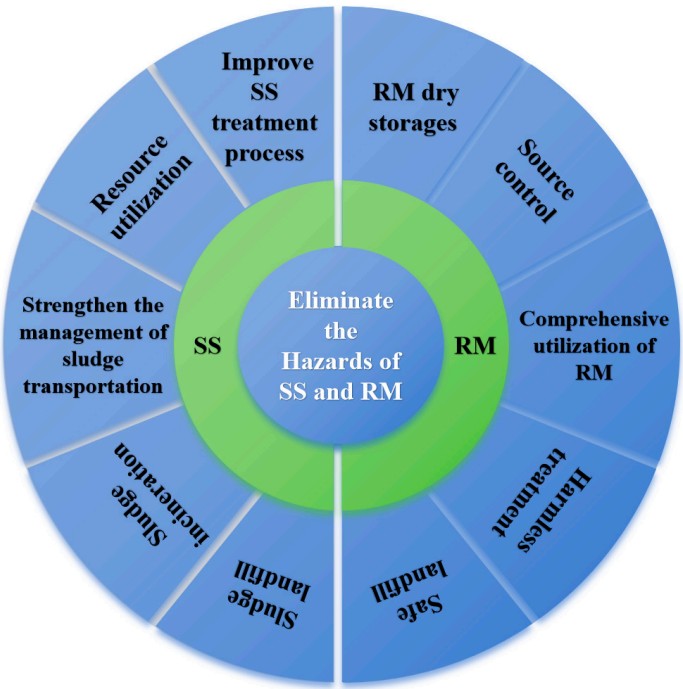

**Figure 2.** Treatment and disposal measures for the elimination of SS and RM hazards [25,35].

In recent years, a large number of research papers have proposed a composite phase change TES technology that uses skeleton materials (SMs) to encapsulate phase change materials (PCMs); the PCMs are wrapped in the pores of the SMs to avoid leakage. $Al_2O_3$ [36,37], MgO [38,39], $Ca(OH)_2$ [40], expanded graphite [41,42], expanded vermiculite [43,44], tailings [45,46], SiC [47,48], semi-coke ash [49,50], fly ash [51–53], and kaolinite [54,55] are all used as SMs to prepare energy storage composites (ESCs) for TES. SS and RM can also be used as SMs after pretreatment and combined with PCMs to prepare ESCs.

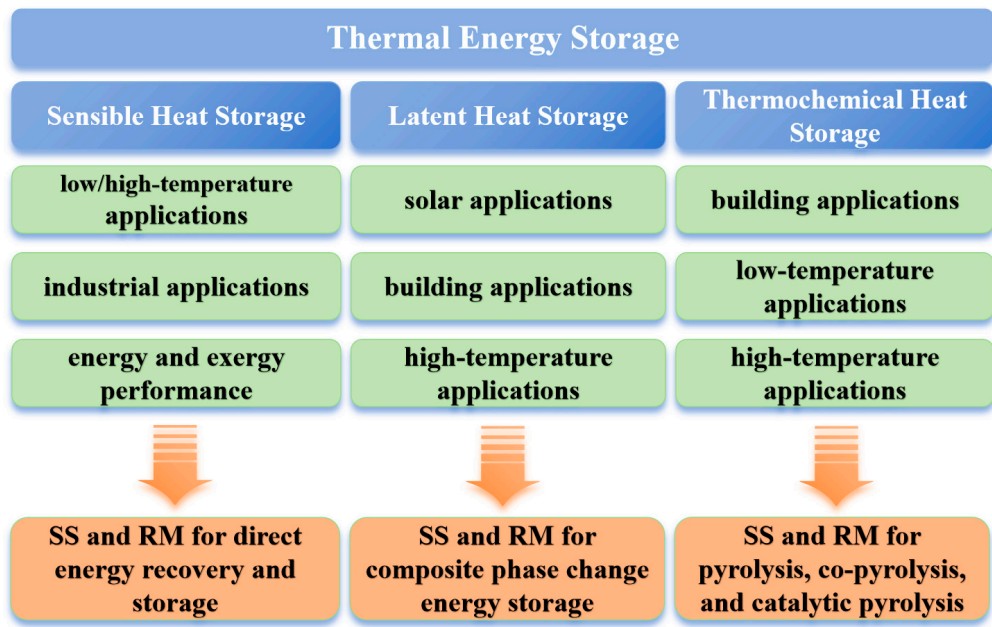

**Figure 3.** Applications of TES technologies [32,56–60].

Liu et al. [61] used RM produced during the process of extracting alumina with the Bayer method as the SM and paraffin as the PCM to prepare paraffin–RM ESCs for low-temperature TES applications using the mixed mill-heating method, and concluded that the composite material had an LH of up to 40 J/g in the temperature range of 75–85 °C, with good thermal properties. Xiong et al. [62] used SS-incineration ash as the SM and sodium nitrate as the PCM to prepare SS-$NaNO_3$ ESCs for the heating of buildings and TES by the cold compression–hot sintering method (CCHS method); the results show that the composites have a good thermal stability, and the composites have a best energy storage density of 409.25 kJ/kg when the ratio of SM to PCM is 5:5.

Moreover, Li et al. [63] extracted the humic acid from SS and applied it to supercapacitor material, and the results illustrate that the biogas energy can be efficiently recovered. Studies show that SS and RM can be used for TES, which can reduce environmental pollution and help to realize the recycling of resources [61–63]. The reuse of industrial solid waste in the TES process not only reduces dependence on virgin resources but also reduces the cost of waste disposal, which helps to promote sustainable development and environmental protection [64].

Both SS and RM are waste products generated during industrial production processes, sharing similar treatment requirements and potential utilization values. Against the backdrop of environmental protection and resource recycling, the recycling and utilization of these two waste types has become a research focus. Therefore, discussing SS and RM in the same general review paper is conducive to a comprehensive and systematic analysis of their current recycling status, technological methods, and future development directions, which provides a useful reference and insight for research in the field of TES (Figure 4 illustrates the methodology of the review).

Through this mini review, we hope to provide a useful reference and enlightenment for the development of SS and RM in TES applications. At the same time, it is also expected to inspire more researchers and enterprises to pay attention to this field and jointly promote the technological progress and market promotion of SS and RM in TES applications.

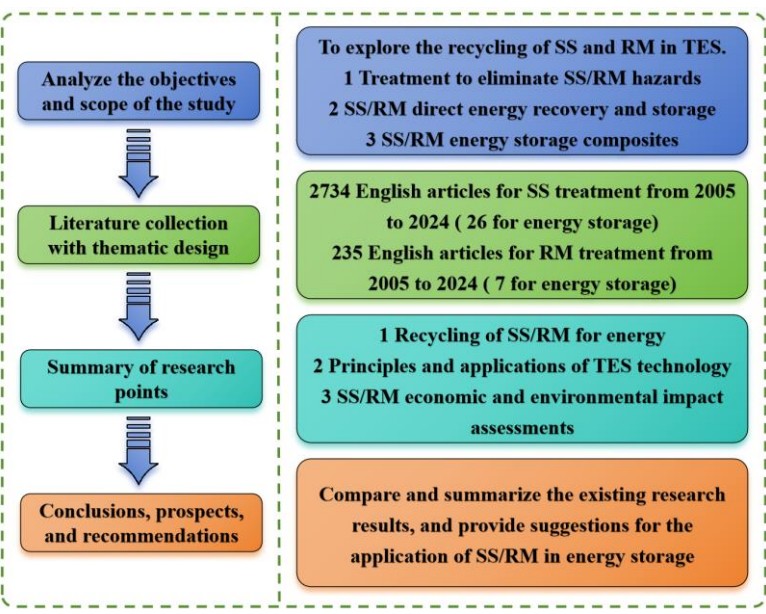

**Figure 4.** Methodological flow chart of the review.

## 2. Basic Property, Recycling Options, and Challenges

### 2.1. Basic Property

2.1.1. Sewage Sludge

Sludge is a solid precipitate produced in the process of water and sewage treatment, which is an extremely complex heterogeneous body, composed of organic residues, bacterial bacteria, inorganic particles, colloids, etc. [15]. The main characteristics of sludge are a high moisture content, high organic matter content (accounting for 60~80%), susceptibility to decay and odor, fine particles (usually between 0.02~0.2 mm), small specific gravity, and gelatinous liquid [5]. In addition, the sludge also contains a certain number of inorganic substances, but the relative proportion is small. According to the different treatment methods and separation processes, sludge can be divided into primary sludge, activated sludge, humus sludge, and chemical sludge [65,66]. Figure 5 illustrates the economic and environmental benefits of different sludges and their feasibility as energy storage materials. The main oxides in SS are $SiO_2$ (10–25%), $Al_2O_3$ (5–10%), and $CaO$ (10–30%), which increase to 25–50%, 10–20%, and 15–30%, respectively, in sludge ash after incineration [67]. Table 1 illustrates the main chemical composition of SS-incineration ash. As the moisture content of SS decreases, the SS form undergoes a change process from pure liquid to viscous, plastic, semi-dry solid, and pure solid. In other words, when the moisture content of SS exceeds 85%, SS presents a fluid state, similar to a liquid, but when the moisture content of SS is 65~85%, the form of SS will change to a plastic state, which is a soft state and not easy to make flow, and when the moisture content is further reduced to less than 60%, SS will be solid, and the SS at this time is relatively dry, stable in shape, and convenient for transportation and treatment [68,69].

Domestic wastewater and industrial wastewater need to be treated through a primary sedimentation tank and a secondary sedimentation tank, and the sewage treatment process and sludge disposal are shown in Figure 6.

It is worth noting that, when SS contains industrial wastewater sludge, it will bring a series of differences in terms of composition, treatment difficulty, environmental impact, and utilization value [70,71]. SS with industrial wastewater sludge may contain more complex components such as heavy metals, inorganic salts, organic matter, grease, etc., which makes the mixed sludge require more professional and refined methods in treatment and disposal [72]. In addition, mixed sludge containing industrial wastewater sludge may be more toxic or hazardous, which increases the difficulty of sludge treatment and may pose a greater potential risk to the environment [73].

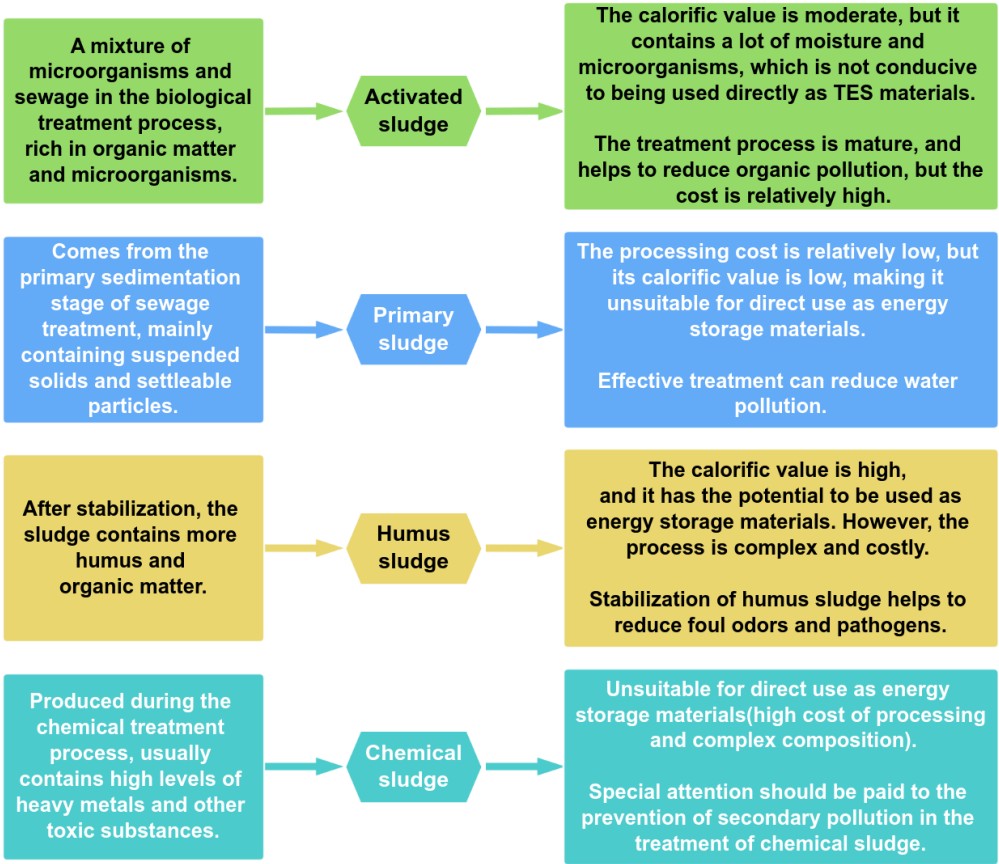

**Figure 5.** Comparisons of different types of sludge [74–80].

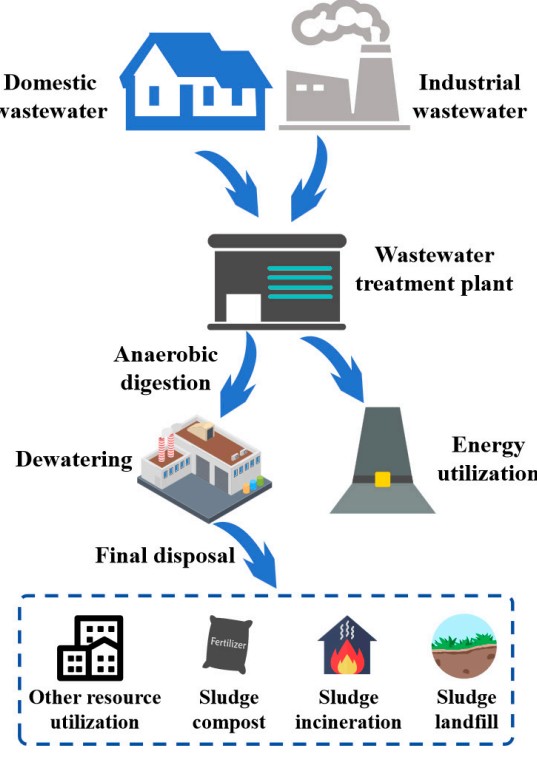

**Figure 6.** Sewage treatment process and sludge disposal, reproduced from [81].

**Table 1.** Main chemical composition of SS-incineration ash.

| Ref. | SiO$_2$ | Al$_2$O$_3$ | P$_2$O$_5$ | CaO | Fe$_2$O$_3$ | MgO | K$_2$O | Na$_2$O | SO$_3$ |
|------|---------|-------------|------------|-----|-------------|-----|--------|---------|--------|
| [62] | 24.95% | 37.04% | 17.31% | 7.75% | 5.75% | 2.59% | 1.69% | 1.57% | 0.07% |
| [82] | 10.30% | 5.04% | 2.42% | 34.10% | 15.40% | 1.14% | 0.45% | 1.27% | 27.70% |
| [83] | 27.27% | 24.21% | 16.97% | 6.95% | 10.22% | 2.45% | 2.07% | 0.32% | |
| [84] | 30.22% | 16.82% | 13.05% | 8.31% | 22.24% | 3.25% | 1.88% | | 3.08% |
| [84] | 43.95% | 17.98% | 12.09% | 6.16% | 12.48% | 2.65% | 2.42% | | 1.54% |

Briefly, the basic characteristics of SS are mainly reflected in its high organic matter content, fine particles, small density, high moisture content, and being not easy to dewater, which determines that the treatment and disposal of sludge need to consider its dewatering performance and possible environmental problems.

### 2.1.2. Red Mud

Red mud (RM) is an iron-rich (about 42%) industrial waste residue that has a red color due to its high content of iron oxide [85]. This waste residue is mainly derived from the process of extracting alumina from bauxite, and is the most important by-product of the production of alumina by sintering, Bayer, and combined methods, and the annual production of red mud is estimated to reach 150 million tons [86]. The main components of RM are SiO$_2$, Al$_2$O$_3$, CaO, Fe$_2$O$_3$, etc., which is an insoluble residue [87,88]. RM has a high water content and contains alkali, heavy metals, and other harmful components; the pH value can reach more than 11, with strong alkalinity, causing a serious impact on the environment and ecology [22,89,90]. The high soluble alkalinity of red mud makes it difficult to sinter into a strong building material, which is the biggest obstacle to its recycling [91]. The hazards caused by RM are multifaceted, including land occupation, air pollution, soil pollution, building erosion, and groundwater pollution [85]. Therefore, the treatment and disposal of RM need to be highly valued, and scientific and effective measures should be taken to reduce its impact on the environment and human health.

The basic principle of the Bayer method for producing alumina is to dissolve the aluminum oxide in the bauxite with sodium hydroxide solution to produce sodium aluminate, and then precipitate the aluminum hydroxide through dilution and cooling, and the specific process flow of the traditional Bayer method to produce alumina is shown in Figure 7. Sintered RM, Bayer RM, and combined RM are three common RM preparation methods, and there are some differences in their preparation process, material properties, and TES performance, as shown in Table 2. The chemical composition of sintered red mud, Bayer red mud, and combined RM is shown in Table 3. The differences in the properties of sintered red mud and Bayer RM are shown in Table 4. Sintered RM has a high density and thermal conductivity, which is suitable for TES applications that require high stability and wear resistance. Bayer RM has good plasticity and machinability and is suitable for some fine products. Combined RM is a combination of sintered RM and Bayer RM, which can strike a balance between performance and TES capacity. Choosing the right red mud type depends on the specific application needs and priorities.

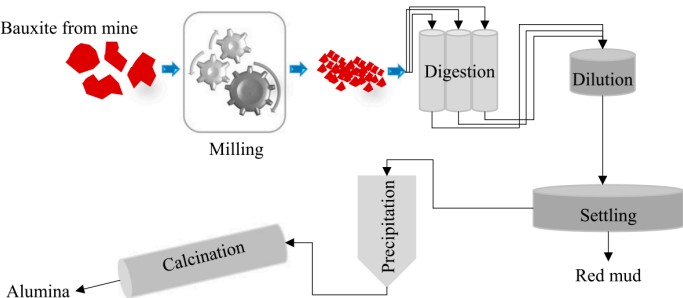

**Figure 7.** The classic Bayer method of RM production [92].

**Table 2.** The differences in the preparation process, material properties and TES performance of sintered RM, Bayer RM, and combined RM [25,93].

| Distinguish | Sintered RM | Bayer RM | Combined RM |
|---|---|---|---|
| Preparation method | Hematite powder is sintered to form solid lumps at high temperatures. | Raw materials reacted with alkaline solutions through chemical reactions, and then prepared by precipitation, filtration, and drying. | By mixing sintered RM and Bayer RM in a certain proportion, which combines the characteristics of the two RMs to obtain better material properties. |
| Material Performance | High stability and mechanical strength; Lower porosity and a higher density; Good thermal conductivity. | High purity and finer particle size; Good plasticity and machinability; Low density and thermal conductivity. | High stability and mechanical strength; Low density and good machinability. |
| Thermal storage performance | With high density and thermal conductivity, can effectively absorb and conduct heat, and release heat quickly. | The TES performance is relatively weak, but can still absorb and release a certain amount of heat. | In terms of heat storage performance, it shows good comprehensive characteristics, with high heat conduction performance and certain heat storage capacity. |

**Table 3.** Main chemical composition of sintered RM, Bayer RM, and combined RM [93,94].

| RM Type | $SiO_2$ | CaO | $Al_2O_3$ | $Fe_2O_3$ | MgO | $Na_2O$ | $K_2O$ | $TiO_2$ | Burn Vectors |
|---|---|---|---|---|---|---|---|---|---|
| Sintered RM | 3~20 | 2.0~8.0 | 10.0~20.0 | 30.0~60.0 | — | 2.0~10.0 | — | 0.01~10 | 10.0~15.0 |
| Bayer RM | 20~23 | 46.0~49.0 | 5.0~7.0 | 7.0~10.0 | 1.2~1.6 | 2.0~2.5 | 0.2~0.4 | 2.5~3.0 | 6.0~10.0 |
| Combined RM | 20~20.5 | 43.7~46.8 | 5.4~7.5 | 6.1~7.5 | — | 2.8~3.0 | 0.5~0.7 | 6.1~7.7 | — |

**Table 4.** Comparisons of the properties and major applications of sintered RM and Bayer RM [95].

| Compare Content | Sintered RM | Bayer RM |
|---|---|---|
| Average particle size (μm) | 28.5 | 14.8 |
| Density (g/cm$^3$) | 3.26 | 2.70 |
| Specific gravity | 2.47 | 2.64 |
| Mass loss in heat treatment (%) | 6.5 | 3.2 |
| Bond strength (kPa) | 287 | 14.6 |
| Stability and water conductivity | Better | Weaker |
| Main applications | Road construction, building materials, cement production, environmental protection, and desulfurization | The main filling material for the dam body construction of the storage yard, the production of cement, the manufacture of ceramics and refractories. |

*2.2. Recycling Options*

The recycling of SS and RM is mainly based on its characteristics, the feasibility of the treatment technology, the economic cost, and the environmental benefits. SS and RM recycling usually involves chemical treatment, reduction, and resource utilization to convert it into valuable products such as fertilizer and energy.

(1) Organic matter transformation and soil improvement The use of microorganisms to degrade and convert organic waste from SS and RM into organic matter can be used for soil improvement, to improve soil fertility, and reduce carbon emissions [96,97]

(2) Carbon capture and utilization The introduction of carbon capture technology to capture and utilize the carbon dioxide produced during sludge treatment, for example, to produce synthetic materials, chemicals, or carbon sequestration [5,16].

(3) Circular economy and resource recovery Recycling useful components in sludge, such as organic matter, phosphorus, metals, etc., which can be used to produce organic fertilizers and chemicals or be recycled to maximize the use of resources [98–100].

(4) Intelligent monitoring and optimal management Using advanced monitoring technology, big data analysis, and artificial intelligence, the sludge treatment process can be intelligently monitored and optimized to improve treatment efficiency and reduce carbon emissions [101].

(5) Energy storage applications The organic components in SS and RM can be used as excellent TES materials after specific treatment [102]. This application method not only improves energy efficiency, but also further reduces carbon emissions in the sludge treatment process, opening up a new means of carbon emission reduction and the utilization of SS and RM.

Compared with other recycling options, the recycling of SS and RM has significant advantages in terms of TES, as they can not only efficiently convert waste into renewable energy and reduce the environmental load, but they also show a broad technical potential and application prospects, with a significance worthy of in-depth study.

*2.3. Challenges*

Song et al. [103] reused SS and RM as a resource, using RM as the oxygen carrier, SS as the fuel, and ultra-low concentration methane as the oxidizing gas, to realize the comprehensive treatment of gas–solid waste. However, due to the low calorific value of pure sludge, it is not economical to use it as a fuel. In future work, mixing it with coal may be considered. A series of problems, such as the blending ratio, temperature, and pollutants, should be systematically studied. Wei et al. [104] utilized blast furnace dust, alkaline oxygen furnace sludge, steel slag magnetic separation powder, and iron as raw materials for blast furnace ironmaking and the resource utilization of zinc-containing dust and sludge in the blast furnace. Ren et al. [105] used SS combined with coal to produce hydrogen for heat generation and utilization, and made an efficient and clean use of sludge and coal. Capodaglio et al. [106] utilized SS as a raw material to produce methane gas, and biohydrogen from sludge fermentation can be used as a clean fuel. Hamalainen et al. [107] have studied the hydrothermal carbonization of SS, which shows that the generated biogas can be upgraded by a membrane filtration system, to produce biomethane liquefied into liquefied biogas as a transportation fuel that can be used for energy recovery. The following are the challenges for SS and RM in terms of energy storage:

(1) Technical problems Although some progress has been made in sludge treatment, carbon emission reduction, and TES technologies, there are still technical bottlenecks. The current technology still needs to be improved in terms of treatment efficiency, energy consumption, and secondary pollution control [108,109].

(2) Treatment cost The treatment cost of SS and RM is high, including the pretreatment, transportation, storage, and subsequent utilization, which increases the economic burden of TES technology's application [110–112]. In 2021, the cost of SS collection and treatment services in cities around the world varied significantly, with costs ranging from as low as 0.1 USD/m$^3$ to as high as 16 USD/m$^3$ [113]. RM costs 11.08 USD/ton, and the accumulation and storage of RM comes with a significant maintenance fee [7].

(3) Environmental protection standards When handling and using these substances, environmental protection standards must be strictly followed to prevent secondary pollution to the environment. Therefore, effective measures need to be taken to control the emission of pollutants in TES applications [114].

(4) Market acceptance As the TES technology of SS and RM is relatively new, the market acceptance of it may not be high, and it is necessary to increase publicity efforts to improve public awareness and acceptance [106].

SS and RM face multiple challenges in TES applications, such as technical bottlenecks, high treatment costs, and environmental risks, and it is expected that, through technological breakthroughs, policy support, and social consensus, a more efficient and environmentally friendly utilization will be achieved in the future and the development of green energy will be promoted.

Although SS and RM face some challenges in CR and TES applications, their potential for resource utilization and energy efficiency improvement cannot be ignored. In addition, SS and RM have shown potential application prospects in direct energy storage and phase change composite energy storage, providing new ways and solutions for TES and waste resource utilization. With the advancement in technology and policy support, the prospects of these two substances in CR and TES applications will be broader.

## 3. Recycling for Thermal Energy Storage

### 3.1. Sewage Sludge

#### 3.1.1. Direct Energy Recovery and Storage

SS is rich in organic matter, which can be treated by a series of technologies, such as incineration, anaerobic digestion, etc., and directly converted into heat or electricity. This conversion process not only effectively utilizes the energy in the sludge and reduces the waste of energy, but also reduces the volume of the sludge and reduces the burden on the environment.

Figure 8 demonstrates the energy recovery route of sludge, which relies on a series of advanced technologies and methods that aim to extract and convert energy from sludge while reducing, stabilizing, and detoxifying the sludge. Sludge incineration is a common method of sludge energy recovery. After the sludge is pretreated, it enters the incinerator for high-temperature incineration, and the heat energy generated during the incineration process can be used to generate electricity or heat [115]. The incineration residue can also be recycled as building materials, to realize the resource utilization of sludge. However, harmful gases may be produced during the incineration process, so emission standards need to be strictly controlled to ensure environmental protection [116]. Bio-gasification technology is a method of using sludge for energy conversion. In a high-temperature and low-oxygen environment, the sludge decomposes organic matter to produce biogas through the action of microorganisms [117]. This biogas is rich in methane and can be used as an alternative to natural gas for cooking and heating. Bio-gasification technology not only realizes the energy utilization of sludge but also reduces greenhouse gas emissions [118]. Anaerobic digestion technology converts the organic matter in sludge into methane and carbon dioxide through microbial degradation [3]. The methane can be further collected and utilized to achieve energy recovery from sludge. At the same time, the stabilized products produced during the anaerobic digestion process can also be used for agricultural fertilizers or soil improvement [119]. Heat pump technology is a method of energy recovery using low-temperature thermal energy. In the sludge treatment process, the low-temperature heat energy contained in the sludge can be extracted by heat pump technology and applied to other fields, such as heating, hot water production, etc. [120]. This technology not only improves energy efficiency, but also reduces dependence on external energy sources.

It should be noted that the selection of a sludge energy recovery route should be comprehensively considered according to the local situation, technical conditions, and economic costs. At the same time, environmental pollution should be strictly controlled in the sludge treatment process to ensure that the treated sludge meets environmental protection standards. With the continuous advancement in science and technology, the energy recovery routes of sludge will be more diversified and efficient in the future.

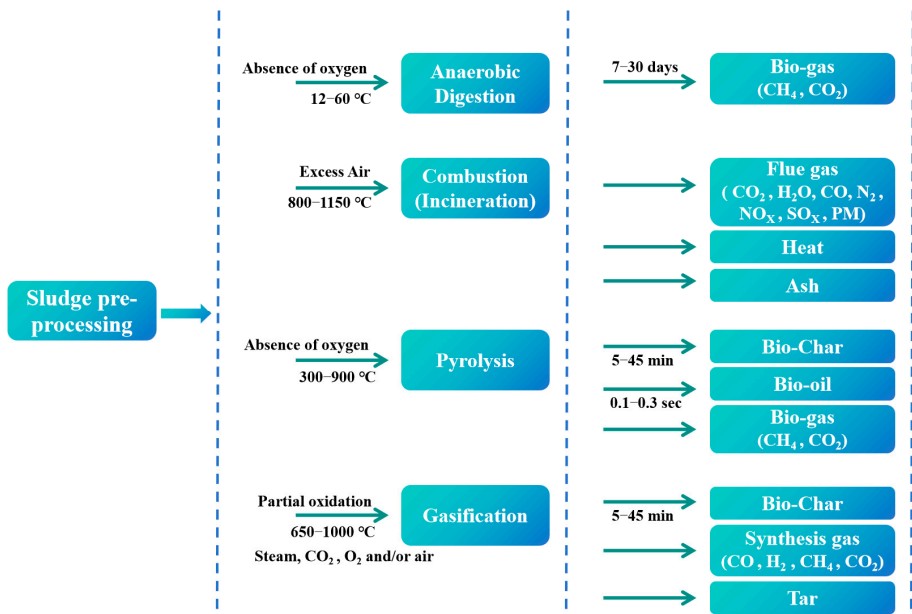

**Figure 8.** Energy recovery of sludge, reproduced from [1].

As shown in Figure 9, SS can be treated not only as an energy material, but also as a contribution to future energy management. Liu et al. [121] used the energy generated by the combustion of sludge and coal for power generation in power plants, as shown in Figure 10, not only turning waste into treasure but also meeting energy needs.

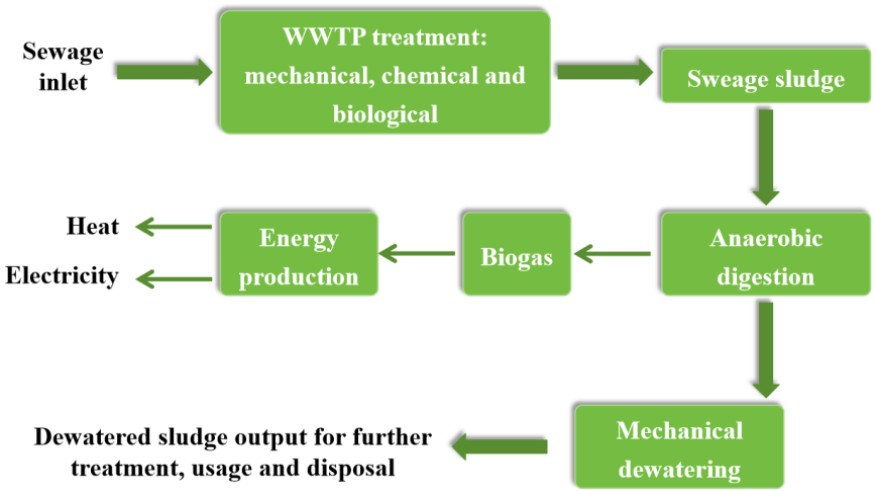

**Figure 9.** Flow diagram of sewage sludge treatment, reproduced from [122].

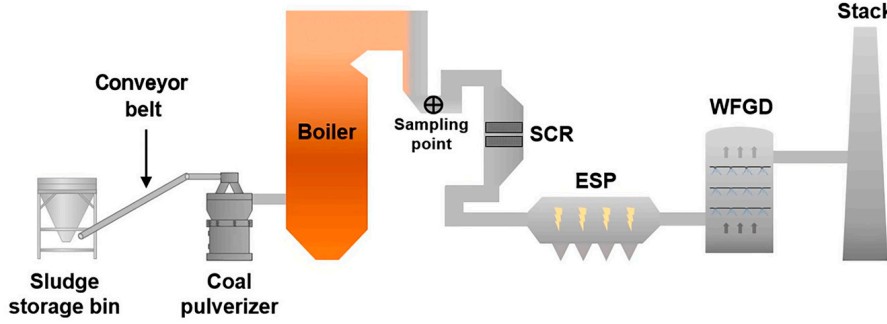

**Figure 10.** Flow diagram of SS for combustion power generation [121].

Meng et al. [123] studied and established a thermal balance model of the sludge pyrolysis–carbonization–cooling–conveying system, pyrolysis–gas incineration–waste heat recovery system, and flue gas treatment and deodorization system, as shown in Figure 11, which can save energy and reduce consumption by 52.2%, compared with the typical sludge drying and incineration process.

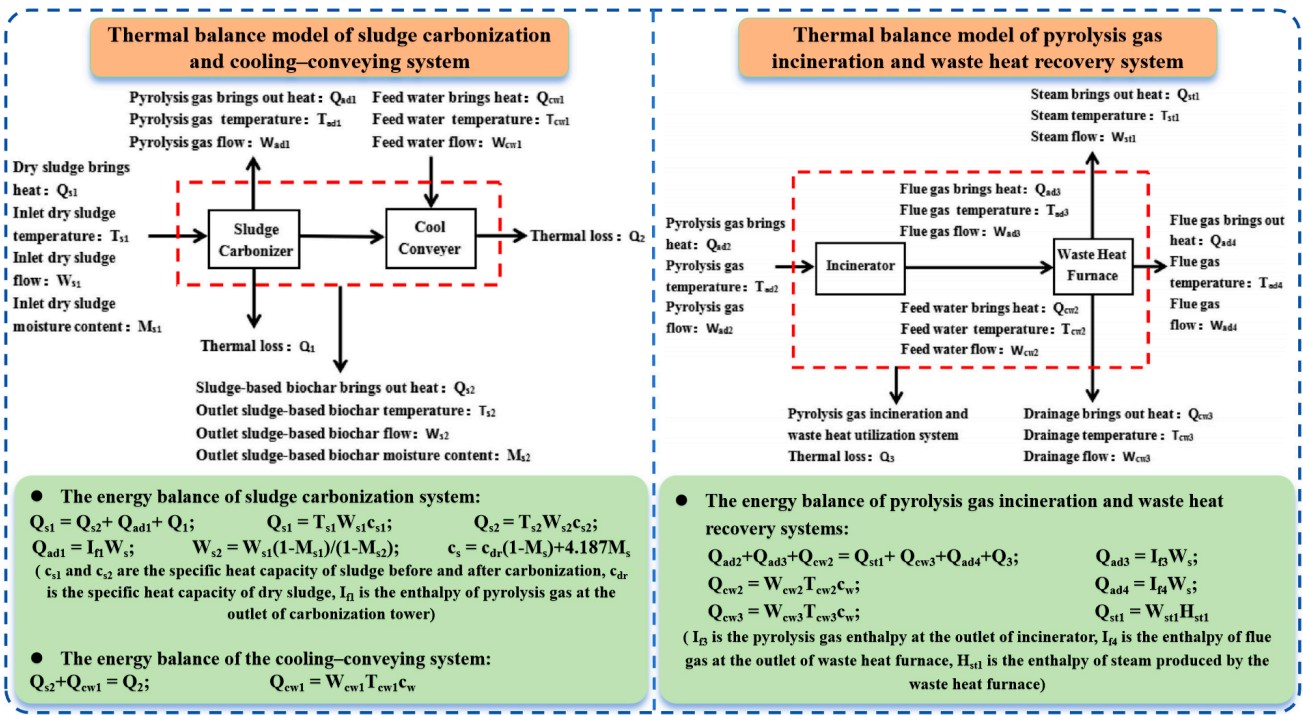

**Figure 11.** Mass and energy balance diagram of sludge carbonization and cooling–conveying system, and pyrolysis gas incineration and waste heat utilization system, reproduced from [123].

Furthermore, the combination of the hydrothermal treatment of SS and coal-fired power generation can improve the combustion efficiency of sludge, while mitigating the hazards associated with co-incineration. Peltola et al. [124] have researched a new type of sludge heat treatment technology that does not require additional energy replenishment when incinerating sludge, and the excess heat generated during the sludge treatment process can also be used for district heating. Hu et al. [125] reviewed the feasibility and challenges of converting sludge into energy materials in the era of carbon neutrality, as well as its potential, and the results show that SS thermochemistry can maximize energy use and material recovery. Liew et al. [126] reviewed the performance of heat treatment technologies in recovering sludge resources, while reducing the environmental impact. Khan et al. [127] carried out a detailed analysis of the production of hydrogen from sludge. In the process of sludge resource utilization, the principles of reduction, stabilization, harmlessness, and resource utilization should be followed.

### 3.1.2. Sewage Sludge Energy Storage Composites

The application of the composite phase change energy storage of SS is an innovative means of energy utilization, aiming to improve energy utilization efficiency and environmental benefits. Composite phase change energy storage technology combines the treatment of SS with the energy storage characteristics of PCMs. In this application, PCMs are used to absorb and release heat energy, while SS is involved in the process as a carrier or medium. PCMs undergo a phase change at a specific temperature (e.g., solid to liquid, or liquid to solid), which is accompanied by the absorption or release of a large amount of heat energy. By incorporating or combining PCMs into SS, the efficient storage and release of heat energy in the sludge can be achieved.

SS ESCs are an innovative TES solution, which combine the resource utilization of SS and the TES characteristics of PCMs, providing a new solution for environmental protection and energy fields. The material is designed to enable the efficient storage and release of heat, while promoting the environmentally friendly treatment and reuse of SS. SS is a solid waste generated in the sewage treatment process, which contains a large amount of organic, inorganic, and microbial matter. Traditional SS treatment methods often face problems such as a high cost, low resource utilization, and environmental pollution [128]. The transformation of it into a composite phase change TES material can not only effectively solve these problems, but also provide a new high-efficiency material for the field of TES. As the core of TES technology, PCMs can undergo a phase transition at a specific temperature, thereby absorbing or releasing a large amount of latent heat. By compounding PCMs with SS, the organic and inorganic components in the sludge can be fully utilized to improve the TES performance and stability of the composite materials.

In the preparation of SS ESCs, it is first necessary to pretreat the SS to remove the impurities and water in it, to obtain dry sludge powder. Then, select the appropriate phase change material and mix it with the sludge powder. During the mixing process, the PCM can be fully combined with the sludge powder by physical or chemical methods to form a homogeneous composite material. As shown in Figure 12, Tian et al. [129] used SS hydrolysis residue modified with sodium dodecylbenzene sulfonate as the SM, and the ESC was prepared by the vacuum impregnation method, with sodium acetate trihydrate as the PCM; the results show that the LH of the ESCs is reduced by about 6% after 100 melting/solidification cycles, and the phase change process maintains a stable TES performance. Han et al. [130] utilized sludge (with a high calcium (Ca) content) for conversion into high value-added photothermal materials (sludge was converted into functional carbon dots), providing a new strategy for waste utilization, as shown in Figure 13; the results show that Ca-doping is a key factor in effectively improving solar heat conversion and thermal stability. Moreover, as mentioned in the introduction, Xiong et al. [62] used SS-incineration ash as the SM and $NaNO_3$ as the PCM to prepare SS-$NaNO_3$ ESCs by the CCHS method, and the fabrication process of SS-$NaNO_3$ ESCs is shown in Figure 14.

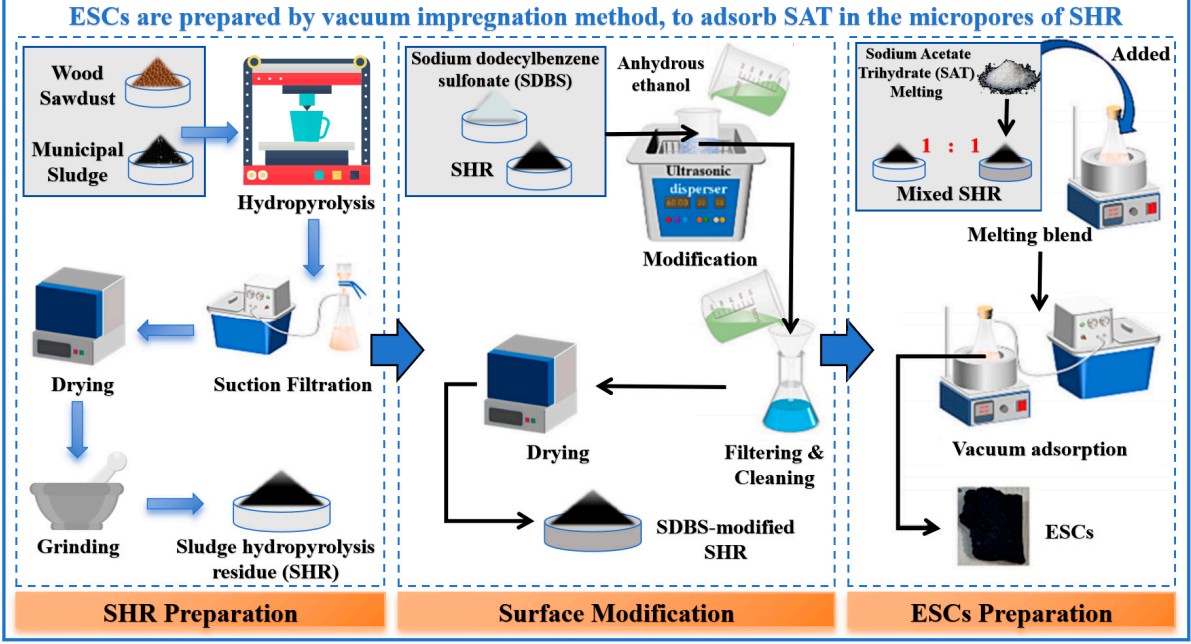

**Figure 12.** Preparation process of SS hydrolysis residue–hydrate salt ESCs, reproduced from [129].

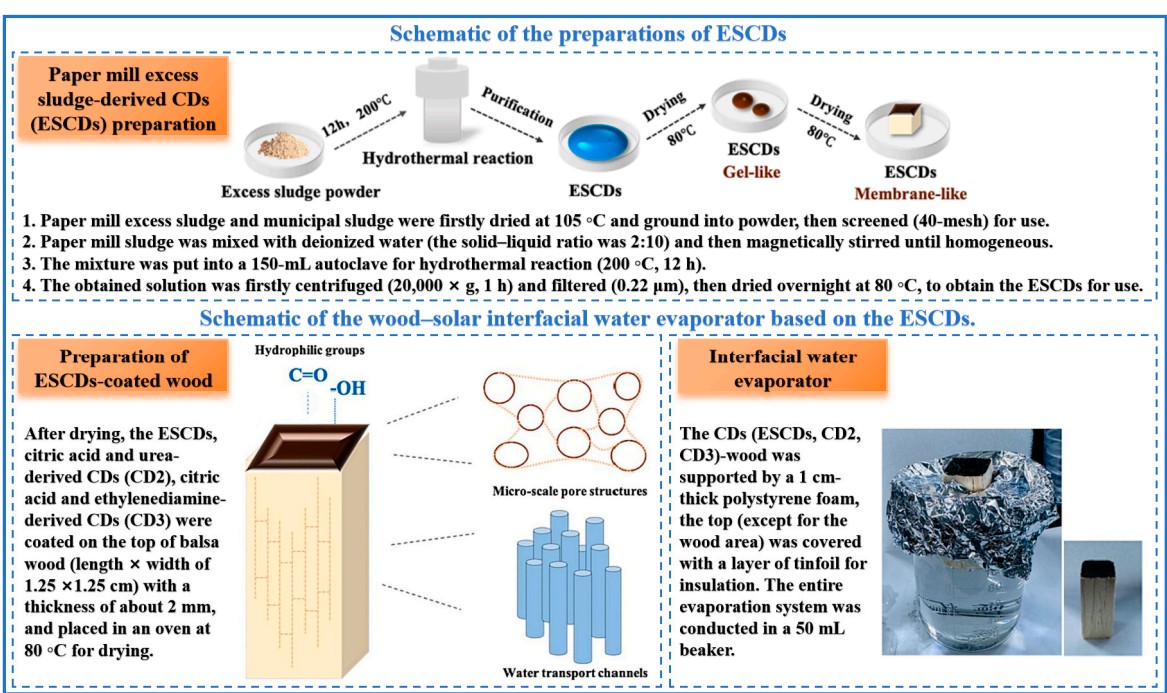

**Figure 13.** Schematic diagram of the preparation of photothermal materials and wood–solar interfacial water evaporator, reproduced from [130].

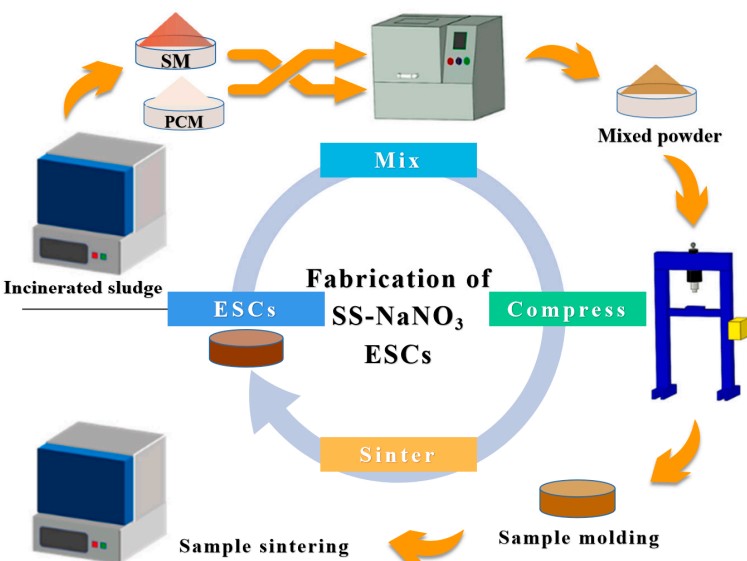

**Figure 14.** Production of SS-NaNO$_3$ ESCs, reproduced from [62].

The application of ESCs not only improves energy efficiency and reduces energy waste but also helps to solve the problem of SS treatment. By combining SS with PCMs, sludge can be reduced, rendered harmless, and resourcefully utilized, and the environmental impact of sludge treatment can be reduced.

However, it should be noted that there are still some technical and economic challenges associated with the application of the composite phase change energy storage of SS. There is a need to develop efficient and stable PCMs and optimize how they are combined with the sludge, while at the same time considering the economy, reliability, and long-term operational performance of the system.

Overall, the application of composite phase change energy storage in SS is a potential means of energy utilization, which can help promote sustainable development and environmental protection.

### 3.2. Red Mud

#### 3.2.1. Direct Thermal Energy Storage

The direct TES application of RM is an area that has gradually attracted attention in recent years, with the increasing awareness of environmental protection and energy utilization. As a strong alkaline solid waste residue produced in the alumina production process, RM contains a large amount of iron oxide and other metal oxides, showing great potential for energy recovery. The combustible components in RM can be converted into heat or electricity through thermochemical conversion technologies such as incineration. This conversion process not only reduces the amount of RM accumulation, but also enables an efficient use of energy. At the same time, the heat generated during the incineration process can be used for heat or power generation, further improving energy efficiency. Microwave-assisted biomass pyrolysis to prepare biofuels, by using high-temperature pretreated RM as additives, increases the energy recovery efficiency from 4.71% to 9.98% [131]. As an additive, RM can significantly optimize the combustion performance of refuse-derived fuels (RDFs). Specifically, when 10% (by mass) of RM is added to the RDF, the combustion index can be increased by up to 91%, and the composite combustion characteristic index can achieve an astonishing increase of 156% [132].

Moreover, Chai et al. [133] utilized bauxite tailings, Bayer RM, and magnetite as raw materials to prepare regenerated thermal-oxidized ceramics with a high energy storage; the results show that the compressive strength of the ceramic is 134.96 MPa, with a low manufacturing cost and high thermal insulation performance. Moreover, as shown in Figure 15, to effectively absorb solar energy and improve the utilization rate of solar collectors, RM-based nanofluids were prepared by Kumar et al. [134], and the results show that RM-based nanofluids are beneficial as cooling fluids, due to their chemical composition, in green energy applications such as solar energy. Yang et al. [135] investigated RM–manganese co-doped high-alumina cement-stabilized carbide slag as a potential candidate material for $CaO/CaCO_3$ thermal storage systems (as shown in Figure 16); the results show that, when RM/Mn are co-doped, the cyclic TES stability of the composites improves with the decrease in RM content.

However, there are still some challenges to the direct TES of red mud. The composition of red mud is complex, containing a variety of metal oxides and impurities, which increases the difficulty and cost of energy recovery. In addition, the alkaline nature of RM may adversely affect the environment, so effective protective measures need to be taken during treatment. There is a need for further research and development of efficient and environmentally friendly energy recovery and storage technologies suitable for the characteristics of RM.

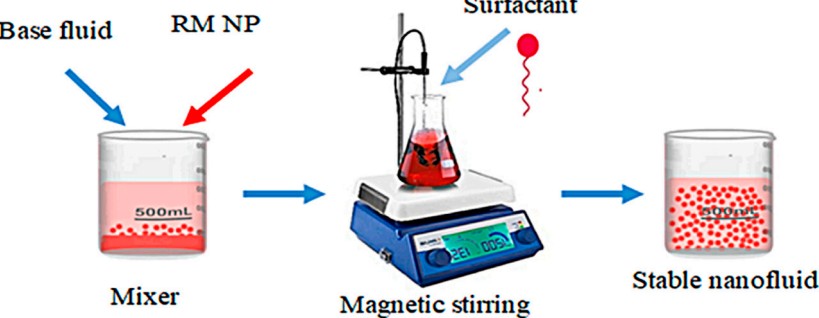

**Figure 15.** Preparation of RM-based nanofluids [134].

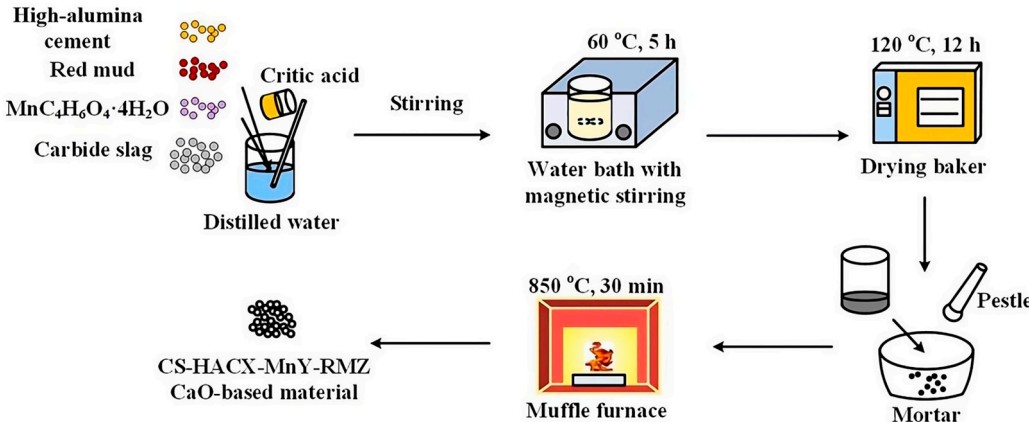

**Figure 16.** Preparation of CaO/CaCO$_3$ ESCs [135].

The direct energy recovery and storage application of RM has great potential and value, which is conducive to the resource utilization of waste and the sustainable development of energy. With the advancement in technology and the improvement of environmental protection requirements, it is believed that this field will receive more attention and investment.

### 3.2.2. Red Mud Energy Storage Composites

RM ESC$_S$ are a new type of TES material that combine the characteristics of RM and PCMs. This material aims to take advantage of the resource advantages of RM and at the same time give full play to the efficient TES performance of PCMs, to achieve the efficient storage and release of heat energy. As a solid waste generated in the process of producing alumina from bauxite, RM is rich in mineral components, such as iron, aluminum, silicon oxides, etc. These components make RM have a high heat capacity and good thermal stability, which is suitable as a matrix for TES materials. PCMs are a class of substances that can undergo a phase transition at a specific temperature, thereby absorbing or releasing a large amount of latent heat. This characteristic makes PCMs have a broad application prospect in the field of TES. By introducing PCMs into red mud, RM ESCs can be prepared to achieve the efficient storage and regulation of thermal energy.

The preparation process of RM ESCs usually includes the pretreatment of RM, the selection and addition of PCMs, and the molding and curing of composite materials. In the preparation process, it is necessary to optimize the proportion of each component and the preparation process to ensure that the composite material has a good TES performance and stability. Afolabi et al. [136] encapsulated paraffin PCM with expanded graphite and compounded it with an RM polymer, using vacuum impregnation technology, to prepare thermal comfort materials for the construction industry, and concluded that the combination of PCM and additives of red mud and an alkaline activator can successfully develop a unique composite material for building walls. Yu et al. [102] used capric acid and paraffin as a PCM (LH value of 143.4 J/g) and micro-encapsulated it with nano-silica and RM, and added it to cement mortar to improve the energy efficiency of buildings, with the conclusion that RM-based ESCs have a good TES capacity (73.5 J/g), chemical compatibility, and thermal stability. Anagnostopoulos et al. [137] used RM as the SM, and solar salt (60% NaNO$_3$ and 40% KNO$_3$) as the PCM, to prepare RM–solar salt ESCs for medium- and high-temperature TES and waste heat recovery and utilization; the results show that the maximum LH of the ESCs is 58 J/g, and the TES density is calculated to be up to 1396 MJ/m$^3$, which demonstrates that the composite material is an ideal TES material. The team also found that the addition of graphite increased the porosity and thermal conductivity of the RM–solar salt ESCs and that the LH of the material decreased as the carbon content increased [12].

Future work may focus on a multi-objective optimization design of the amount of RM-based ESCs in customized cement mortar, to achieve synergistic improvements in the

cost-effectiveness, temperature control capacity, energy saving, and emission reduction performance of buildings.

As RM comes from industrial waste residues, the use of it as a TES material contributes to environmental protection and resource recycling. RM thermal storage material is a potentially environmentally friendly TES material, which is expected to play an important role in heat utilization, energy conservation, and emission reduction in the future.

### 3.3. Analysis and Comparisons

3.3.1. Benefits of Sewage Sludge and Red Mud in Energy Storage

Solid waste SS and RM can be recycled and utilized as resources. Bhattacharya et al. [138] successfully verified the applicability of red mud in the field of energy storage, confirming its potential as an efficient energy storage material. Using SS and RM in energy storage systems offers the following benefits:

(1) Resource reuse: SS and RM, as waste products from industrial production processes, contain certain energy and material components. By applying them to energy storage systems, we can achieve the reuse of these wastes, converting them from a burden into valuable resources [139,140].

(2) Cost reduction: Compared to traditional energy storage materials, SS and RM are less costly as waste products. Using these wastes as energy storage materials can reduce the construction and operating costs of energy storage systems, thus improving economic efficiency [12,62].

(3) Environmental friendliness: The effective utilization of SS and RM helps reduce the discharge and accumulation of these wastes, thereby reducing their potential impact on the environment. By using them in energy storage systems, we can reduce the exploitation and consumption of natural resources, mitigating the risks of environmental pollution and ecological destruction [103].

(4) Innovative technology application: The application of SS and RM in energy storage systems represents a technological innovation. This application approach not only contributes to the development and progress of related technologies but also brings new solutions and ideas to the field of energy storage.

It is worth noting that, while the use of SS and RM in energy storage systems offers these benefits, factors such as stability, safety, and the feasibility of processing technologies must also be considered in practical applications. Therefore, sufficient research and testing are required before promoting and applying this technology, to ensure its effectiveness and reliability.

3.3.2. Comparisons of Energy Storage Composites

Both SS and RM can be used as SMs to prepare ESCs, as shown in Table 5; it is clear that sludge-incineration ash slag-based ESCs have a high mechanical strength, large heat storage density, and can operate at medium and high temperatures. However, there are not enough relevant studies at present, and more research on ESCs based on SS and RM needs to be undertaken.

**Table 5.** Comparison of different ESCs.

| ESCs | PCM, wt.% | Operating Temperature Range, °C | Latent Heat, J/g | TES Density, J/g | Thermal Conductivity, W/(m·K) | Mechanical Strength, MPa | Ref. |
|---|---|---|---|---|---|---|---|
| NaNO$_3$/SS incinerated ash | 50 | 100–400 | 60.33 | 409.25 | 0.955 | 139.65 | [62] |
| Solar salt/RM | 50 | 25–400 | 58 | / | 0.77–0.83 | / | [137] |
| Paraffin/RM | 55 | 50–100 | 40 | / | / | / | [61] |
| Solar salt/RM | 60 | 25–500 | 65.47 | / | / | / | [12] |
| NaNO$_3$/Desulfurization gypsum-carbide slag (7:3) | 50 | 100–400 | 77.38 | 483.2 | 1.548 | 134.1 | [141] |
| Myristic acid/Resin | 60 | 25–100 | / | 210.8 | / | 25–34 | [142] |
| NaNO$_3$/Ca(OH)$_2$ | 60 | 140–340 | 102.8 | 417 | / | 108 | [143] |

**Table 5.** *Cont.*

| ESCs | PCM, wt.% | Operating Temperature Range, °C | Latent Heat, J/g | TES Density, J/g | Thermal Conductivity, W/(m·K) | Mechanical Strength, MPa | Ref. |
|---|---|---|---|---|---|---|---|
| NaNO$_3$/Semi-coke ash | 47 | 100–400 | 69.54 | 424.91 | 1.844 | 113.82 | [49] |
| Na$_2$CO$_3$/Carbide slag | 47.5 | 100–900 | 81.10 | 993 | 0.62 | 22.02 | [144] |
| NaNO$_3$/Steel slag-carbide slag (5:5) | 50 | 100–400 | 74.1 | 371.1 | 1.263 | 131.2 | [145] |
| KNO$_3$/expanded vermiculite | 87 | 243.1–325.1 | 83.8 | / | 0.33 | / | [146] |
| Na$_2$CO$_3$/Carbide slag | 50 | 100–400 | 59.61 | 447 | 0.93 | 73.6 | [147] |
| Li$_4$(OH)$_3$Br/MgO | 70 | 40–350 | 149 | / | 0.626 | / | [38] |
| n-octadecane/Kaolinite | 30 | 25–600 | 81.8 | / | / | / | [54] |
| Na$_2$CO$_3$/Semi-coke ash | 47.5 | 100–900 | 62.9 | 961.58 | 1.306 | 23.57 | [148] |

## 4. Conclusions

Exploring the recycling of SS and RM in terms of TES, it is not difficult to see that these two types of waste have significant potential and value, and we can draw the following conclusions:

(1) SS and RM, as solid waste, exhibit significant resource recovery potential in TES technology and carbon reduction. Sludge treatment has the potential to achieve an 83.48% reduction in carbon emissions compared to conventional cement. Microwave-assisted biomass pyrolysis to prepare biofuels, by using high-temperature pretreated RM as additives, increases the energy recovery efficiency from 4.71% to 9.98%.

(2) However, SS and RM also face several challenges, like high collection and treatment cost, environmental risks, and practical application difficulties in TES applications. The cost of SS collection and treatment services in cities around the world varies significantly, with costs ranging from 0.1 USD/m$^3$ to 16 USD/m$^3$. The accumulation and storage of RM comes with a significant maintenance fee, costing 11.08 USD/ton.

(3) The composition of SS and RM is complex, and its physical and chemical properties may affect the binding effect with PCMs. How to select the appropriate PCM, and how to achieve a uniform mixing and close binding with the SM, is the key technology in the preparation process. In addition, factors such as the material stability, thermal conductivity, and phase transition temperature need to be considered during the preparation process, which require in-depth research and precise control.

(4) With the improvement in environmental awareness and the increase in energy demand, the TES application of SS and RM has broad prospects. A thermal balance model of the sludge pyrolysis–carbonization–cooling–conveying system, pyrolysis gas incineration–waste heat recovery system, and flue gas treatment and deodorization system can save energy and reduce consumption by 52.2%, compared with the typical sludge drying and incineration process.

(5) SS and RM have great potential in the field of TES, but also face many challenges. Through continuous exploration and innovation, we can overcome these challenges, promote the development of resource utilization and TES technology, and contribute to environmental protection and energy utilization.

## 5. Future Perspective and Recommendations

Future perspectives:

(1) The focus will be on improving the energy storage capacity and thermal stability of SS/RM-based materials, which involves exploring novel processing techniques and additives that can enhance the material's phase change behavior and thermal conductivity.

(2) There will be a growing interest in developing a cost-effective and environmentally friendly SS/RM energy storage system, which includes the utilization of renewable energy sources for material processing, as well as the implementation of waste reduction and recycling strategies throughout the manufacturing cycle.

(3) The integration of SS/RM-based TES systems into various applications will be explored. Potential areas of application include building energy management, solar energy systems, and industrial heat recovery.

Recommendations:

(1) Encourage further research and development efforts to enhance the performance and scalability of SS/RM-based TES materials, which includes exploring new material formulations, processing techniques, and additives that can improve the material's properties.
(2) Promote the development of sustainable production processes that minimize environmental impact and maximize resource utilization. This includes the use of renewable energy for material processing, waste reduction strategies, and recycling initiatives.
(3) Foster collaboration between research institutions, industries, and policymakers to accelerate the development and deployment of SS/RM-based thermal energy storage solutions, which includes sharing knowledge, resources, and best practices to overcome challenges and achieve common goals.

**Author Contributions:** Y.X.: Conceptualization, Methodology, and Writing—review and editing. A.Z.: Investigation, Extensive literature review, Writing—original draft. Y.Z.: Methodology, Writing—review and editing. Q.X.: Writing—review and editing. Y.D.: Conceptualization and Methodology. All authors have read and agreed to the published version of the manuscript.

**Funding:** This research was funded by the Scientific Research Program of the Beijing Municipal Education Commission (grant number: KM201910016011), Beijing University of Civil Engineering and Architecture Doctoral Research Start-up Fund (grant number: Z13086) and the National Natural Science Foundation of China (grant number: 52006008). The authors are grateful for the financial supports.

**Conflicts of Interest:** The authors declare no conflict of interest.

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
