# Peer review of "A Mini Review on Sewage Sludge and Red Mud Recycling for Thermal Energy Storage"

_energies, doi:10.3390/en17092079_

Round 1

Reviewer 1 Report

Comments and Suggestions for Authors

In this contribution, the authors summarized the development of utilization of industrial waste sludge and the remaining challenges due to the difficulties from complex composition and high-cost issues. Specifically, organics could be transformed from sludge using gasification, pyrolysis, and anaerobic digestion. In addition, the use of sludge in the field of thermal energy storage was discussed in detail. The manuscript was well written and introduced the sewage sludge recycling development, however, this manuscript cannot be published at its current stage, here are some questions that need to be addressed first.

1) The introduction part is not informative enough. More content about the treatment for industrial sludge should be added. What about the development of this field for the economy or environment so far?

2) Also, some sentences missed relative references. Like:

…which has significant environmental protection and economic benefits.

…requires the resource utilization 42 of various industrial sludges.

3) The authors should take seriously Figure organization; it should be precise and clear.

Figure 5 is hard to follow.

Same question for Figures 6,10 and 14.

4) A section about the future perspective for this field should be added.

Author Response

Dear Reviewer,

Thank you for your thorough and valuable feedback on our manuscript. We have carefully considered each of your comments and have provided a point-by-point response in the Word file as follows.

Thank you very much! 

Reviewer 2 Report

Comments and Suggestions for Authors

Good Day! Look at the file, please

Author Response

Dear Reviewer,

We appreciate your time and effort in reviewing our manuscript and providing valuable feedback to improve the quality of the paper. We have carefully considered each of your comments and have provided a point-by-point response in the Word file as follows.

Thank you again for your comments and support.

Best regards,

Reviewer 3 Report

Comments and Suggestions for Authors

A mini-review on sewage sludge and red mud recycling for thermal energy storage. The resource utilization of industrial waste sludge is particularly important to carbon emission reduction for reaching carbon peak and carbon neutrality. Through anaerobic digestion, pyrolysis gasification, and other technical means, the organic components in the sludge are transformed, which effectively reduces greenhouse gas emissions. Meanwhile, the sludge contains high organic components, which can be used as an excellent energy storage material after treatment. The purpose of this review is to provide a reference for the research and practice of carbon emission reductions and utilization of industrial sludge in thermal energy storage and to promote the continuous development of its in-depth research and application. Below, I have outlined constructive comments aimed at enhancing the professionalism and comprehensiveness of your work. I encourage you to consider these suggestions carefully to ensure your study reaches its full potential.

1.      In the introduction mention the hazards of RM and SW and how they could be eliminated.

2.      For Fig.2 it would be better to not write abbreviations as TES, SHS, or LHS.

3.      There is a lack of literature about the benefits of using red mud or sludge in storage systems. For example, in thermal desalination systems, the energy storage materials could increase the harvested water. Please support with previous results.

4.      For sewage sludge, there are various types as activated sludge and primary sludge. Comparison between their properties should be mentioned in the table. Also, which type is preferred as an energy storage material?

5.      What is the difference if the sewage sludge contains industrial wastewater sludge?

6.      In part 2.3. Challenges and prospects. The economic issue must be discussed when it decided to be used as an energy storage material

7.      Compare red mud, sewage sludge, and other energy storage materials such as paraffin wax, stones, and black painted metals in thermal conductivity and thermal capacity.

8.      After the Conclusion part, it would be better to write A recommendation part.

9.      I think that the carbon capture paragraph is not suitable for the title of the paper.

10.  The paper needs a suitable comparison table that contains various previous studies about RM and SS utilized as an energy storage material. The amount of material used, the final efficiency improvement, etc.

Comments on the Quality of English Language

There are some typos in the manuscript, read thoroughly and improve readability of the manuscript.

Author Response

Dear Reviewer,

Thank you for your detailed comments and suggestions on our manuscript. We have carefully read and considered each of your points, and we appreciate the time and effort you have invested in providing feedback.

In response to your comments, we have made the Word file as follows.

Thank you again for your valuable comments and support.

Best regards,

Round 2

Reviewer 1 Report

Comments and Suggestions for Authors

This revised manuscript provides a comprehensive summary of the progress made in using industrial waste sludge, while also highlighting the continuing hurdles posed by its complicated composition and high-cost concerns. Additionally, it offers valuable insights into the future prospects of this sector. Therefore, it is appropriate for publishing in the Energies journal.

Reviewer 2 Report

Comments and Suggestions for Authors

The manuscript was significantly processed.

The main comments are fixed.

Reviewer 3 Report

Comments and Suggestions for Authors

The authors answered all the questions suggested by the referees.